# β-Catenin Regulates Wound Healing and IL-6 Expression in Activated Human Astrocytes

**DOI:** 10.3390/biomedicines8110479

**Published:** 2020-11-06

**Authors:** Venkata Viswanadh Edara, Shruthi Nooka, Jessica Proulx, Satomi Stacy, Anuja Ghorpade, Kathleen Borgmann

**Affiliations:** 1Department of Pharmacology and Neuroscience, University of North Texas Health Science Center, Fort Worth, TX 76107, USA; venkata.viswana.chowdary.edara@emory.edu (V.V.E.); JessicaProulx@my.unthsc.edu (J.P.); Satomi.Stacy@unthsc.edu (S.S.); 2Department of Microbiology, Immunology and Genetics, University of North Texas Health Science Center, Fort Worth, TX 76107, USA; shruthi.nooka@utsouthwestern.edu (S.N.); anuja.g.ghorpade@gmail.com (A.G.)

**Keywords:** astroglia, HIV-associated neurocognitive disorders (HAND), IL-6 regulation, Wnt/β-catenin signaling, neuroinflammation, NF-κB crosstalk

## Abstract

Reactive astrogliosis is prominent in most neurodegenerative disorders and is often associated with neuroinflammation. The molecular mechanisms regulating astrocyte-linked neuropathogenesis during injury, aging and human immunodeficiency virus (HIV)-associated neurocognitive disorders (HAND) are not fully understood. In this study, we investigated the implications of the wingless type (Wnt)/β-catenin signaling pathway in regulating astrocyte function during gliosis. First, we identified that HIV-associated inflammatory cytokines, interleukin (IL)-1β and tumor necrosis factor (TNF)-α induced mediators of the Wnt/β-catenin pathway including β-catenin and lymphoid enhancer-binding factor (LEF)-1 expression in astrocytes. Next, we investigated the regulatory role of β-catenin on primary aspects of reactive astrogliosis, including proliferation, migration and proinflammatory responses, such as IL-6. Knockdown of β-catenin impaired astrocyte proliferation and migration as shown by reduced cyclin-D1 levels, bromodeoxyuridine incorporation and wound healing. HIV-associated cytokines, IL-1β alone and in combination with TNF-α, strongly induced the expression of proinflammatory cytokines including C-C motif chemokine ligand (CCL)2, C-X-C motif chemokine ligand (CXCL)8 and IL-6; however, only IL-6 levels were regulated by β-catenin as demonstrated by knockdown and pharmacological stabilization. In this context, IL-6 levels were negatively regulated by β-catenin. To better understand this relationship, we examined the crossroads between β-catenin and nuclear factor (NF)-κB pathways. While NF-κB expression was significantly increased by IL-1β and TNF-α, NF-κB levels were not affected by β-catenin knockdown. IL-1β treatment significantly increased glycogen synthase kinase (GSK)-3β phosphorylation, which inhibits β-catenin degradation. Further, pharmacological inhibition of GSK-3β increased nuclear translocation of both β-catenin and NF-κB p65 into the nucleus in the absence of any other inflammatory stimuli. HIV+ human astrocytes show increased IL-6, β-catenin and NF-κB expression levels and are interconnected by regulatory associations during HAND. In summary, our study demonstrates that HIV-associated inflammation increases β-catenin pathway mediators to augment activated astrocyte responses including migration and proliferation, while mitigating IL-6 expression. These findings suggest that β-catenin plays an anti-inflammatory role in activated human astrocytes during neuroinflammatory pathologies, such as HAND.

## 1. Introduction

More than 30% of human immunodeficiency virus-1 (HIV-1) positive patients develop disorders associated with the central nervous system (CNS) [1,2]. Despite effective antiretroviral therapy (ART), HIV-1 invades the brain and causes neurological impairment, commonly referred to as HIV-associated neurocognitive disorders (HAND). Even in the post-ART era, low-level viral replication, HIV-1-associated neuroinflammation and oxidative stress contribute to neuropathogenesis [3,4], which highlights the significant need for identifying indirect host-regulated mechanisms for HAND adjunctive therapy.

Astrocytes, the most abundant glial cell in the brain, maintain CNS homeostasis and blood-brain barrier (BBB) integrity [5,6]. Reactive astrogliosis is a process where astrocytes respond to CNS insults, including infection, trauma, ischemia and neurodegeneration. Reactive astrocytes secrete a wide variety of pro- and anti-inflammatory factors that can potentially mediate neuroprotective or neurotoxic effects depending upon the severity, type, context and duration of the insult [7,8,9]. The severity of HAND is strongly associated with astrocyte dysfunction and immune activation. Inflammation mediates CNS damage and is a major contributor to HAND in HIV-infected individuals. Furthermore, HIV-1 infected microglia and perivascular macrophages, release various inflammatory and neurotoxic factors [10,11]. Several inflammatory cytokines including tumor necrosis factor (TNF)-α, interleukin (IL)-1β, and IL-6 are elevated in the CNS or cerebrospinal fluid of HAND individuals [10,12,13,14]. Multiple studies have shown upregulation of inflammatory cytokines such as C-X-C motif chemokine ligand (CXCL)8, C-C motif chemokine ligand (CCL)2, CCL3, interferon (IFN)-γ and IL-6 production by astrocytes in response to macrophage-derived IL-1β and TNF-α during HIV-1 CNS infection [15,16,17,18]. These cytokines indirectly mediate their neurotoxic effects by activating astrocytes and play an important role in the further induction of neuronal injury and HAND pathogenesis [19,20,21,22].

The canonical wingless type Wnt (Wnt/β-catenin) signaling pathway is an important intracellular signal transduction pathway that promotes cell survival, differentiation, proliferation and communication. It is implicated in several neurodegenerative diseases including Alzheimer’s disease, Parkinson’s disease and neurological complications of acquired immunodeficiency syndrome [23,24,25,26]. Wnt/β-catenin pathway signaling restricts the HIV life cycle in target cell types, including peripheral blood mononuclear cells (PBMC) and astrocytes [27,28]. Wnt ligands are specifically upregulated in spinal cord dorsal horns of HIV+ patients with chronic pain, suggesting a critical role in pathogenesis of HIV-associated pain [29,30]. A hallmark of the Wnt/β-catenin pathway is the stabilization of cytosolic β-catenin, which translocates to the nucleus and activates the transcription of Wnt target genes [31]. Glycogen synthase kinase (GSK)-3β is highly active, which phosphorylates and degrades β-catenin through proteosomal degradation. GSK-3β has constitutive kinase activity, which is often negatively regulated through posttranslational modifications. In contrast, inactivation of GSK-3β activity inhibits β-catenin degradation. GSK-3β, as an upstream regulator of β-catenin, has been shown to play a major role in neuroinflammation in several neurodegenerative diseases [32,33,34]. 

The nuclear factor (NF)-κB signaling cascade is a pivotal modulator of inflammation and regulates innate and adaptive immune responses, cell proliferation, and apoptosis in various cell types [35,36]. While basal NF-κB activity is generally low in astrocytes, it increases significantly during inflammation, injury or disease [37,38,39,40]. Crosstalk between Wnt/β-catenin and NF-κB signaling during inflammation in various peripheral cell types was well summarized by Ma et al. [41]; however, little is known about the contribution of this crosstalk to the neuroinflammatory responses during astrogliosis. In the present study, we investigated the regulatory mechanisms mediated by Wnt/β-catenin signaling in the neuroinflammation-associated reactive astrogliosis process.

## 2. Experimental Section

### 2.1. Isolation, Culture and Treatment of Human Astrocytes

Human astrocytes were isolated from brain tissues recovered from first- and early second-trimester conceptual tissue as described previously [42,43,44]. Tissue samples were obtained from the Birth Defects Research Laboratory at the University of Washington in Seattle in full compliance with the ethical guidelines of the National Institutes of Health (NIH), Universities of Washington and North Texas Health Science Center. Our study involved the generation of human neural cells, hence it was reviewed by the North Texas Regional IRB and was found to be exempt under the provision of Title 45 of the Code of Federal Regulations part 46. 104 (d) (ii) wherein the biospecimens were de-identified and the researchers had no contact with subjects and could not re-identify the biospecimens. Isolated astrocytes were cultured in 1:1 v/v DMEM:F12 medium containing 10% fetal bovine serum (catalog no. PS-FB1 lot no. 27C1171, Peak Serum, Wellington, CO, USA) (catalog no. F-05000-D, lot no. F12C18D1, Atlas Biologicals, Fort Collins, CO, USA), penicillin 50 U-streptomycin 50 µg/mL-neomycin 100 μg/mL (PSN) (catalog no. P4083, Sigma Aldrich, St. Louis, MO, USA) and amphotericin B 2.5 μg/mL (catalog no. A2942, Sigma Aldrich). Astrocytes were plated either in 6-well plates or 150 cm^2^ flasks at 2 × 10^6^ cells/well and 20 × 10^6^ cells/flask, respectively, and treated with IL-1β (20 ng/mL, R&D Systems, Minneapolis, MN, USA), TNF-α (50 ng/mL, R&D Systems) and 6-bromoindirubin-3′-oxime (BIO) (500 nM, Millipore Sigma, Burlington, MA, USA) for 8 or 24 h at 37 °C and 5% CO_2_.

### 2.2. Transfection of Astrocytes

Human astrocytes were transfected either with no siRNA (mock), scrambled non-targeting siRNA (siCon, Dharmacon, Lafayette, CO, USA), or siRNA specific to β-catenin (siβ-catenin) using the Amaxa^TM^ P3 primary cell 96-well nucleofector kit (Lonza, Walkersville, MD, USA). Briefly, 1.6 × 10^6^ astrocytes were suspended in 20 μL nucleofector solution containing mock, siCon or siβ-catenin (200 nM) and transfected using a Nucleofector/Shuttle (Lonza) device. Transfected cells were supplemented with media and incubated for 20 min at 37 °C prior to plating. Cells were then cultured and allowed to recover for 48 h prior to experimental use. The knockdown efficiency of siβ-catenin was verified in all donors and averaged −88.5 ± 3.6% (±SD) as assayed by real-time polymerase chain reaction (RT-PCR) and −46.6 ± 3.9% by Western blot as described below.

### 2.3. In Vitro Scratch Assay

A well-established in vitro wound-healing model to study the migration of astrocytes was used [45]. Briefly, mock, siCon and siβ-catenin transfected human astrocytes were plated at a density of 2 × 10^6^ cells per 6-well tissue culture plate well and grown to confluence for 48 h. The media was then aspirated, and a thin stretch of the confluent monolayer was scraped to create a “scratch or injury” using a sterile 10 μL pipette tip. Fresh astrocyte medium was added, and the wound was allowed to heal. The culture plates were examined periodically at 8, 24, 36 and 48 h and then returned to resume incubation. Live images of migrating cells were obtained using a phase contrast microscope (Zeiss Invertoskop 40C) (Carl Zeiss Microscopy, LLC, Thornwood, NY, USA).

### 2.4. Measures of Cell Viability

Metabolic activity of astrocytes was measured by 3-(4,5-dimethylthiazol-2-yl)-2,5-diphenyl tetrazolium bromide (MTT, Sigma Aldrich) assay as previously described [46]. Briefly, 0.25% MTT was added to astrocytes and incubated for 20 to 45 min at 37 °C. Then the MTT solution was removed, and crystals were dissolved in dimethyl sulfoxide (DMSO, Sigma Aldrich) for 15 min with gentle agitation. Absorbance was assayed at 490 nm in a SpectraMax M5 microplate reader (Molecular Devices, Sunnyvale, CA, USA).

### 2.5. Cell Proliferation/Bromodeoxyuridine (BrdU) Incorporation Assay

Transfected astrocytes were plated in triplicates in 96-well tissue culture plates at 5 × 10^4^ cells/well and allowed to recover overnight. BrdU was added 12 h prior to the measurement of incorporation using BrdU Cell Proliferation Assay Kit (catalog no. 6813, Cell Signaling Technology, Inc., Danvers, MA, USA) as per the manufacturer’s instructions. Data were normalized to mock transfected astrocytes.

### 2.6. Quantification of Proinflammatory Cytokines by ELISA

Cell culture supernatants were collected at specified timepoints after treatment and used to quantify proinflammatory cytokines CXCL8, CCL2, and IL-6 (catalog no. D8000C, DCP00, and D6050, respectively, Quantikine ELISA Kits, R&D Systems). Assays were performed as per the manufacturer’s instructions and normalized to MTT levels.

### 2.7. Real Time Gene Expression Analysis

RNA was isolated using Trizol reagent (Thermo Fisher Scientific, Waltham, MA, USA) as previously described [42] followed by DNA digestion and precipitation. A Nanodrop fluorospectrometer (Thermo Fisher Scientific) was used to assess RNA purity and to quantify total RNA levels. Then, transcripts were made into cDNA as per the manufacturer’s instructions (High Capacity cDNA Reverse Transcription Kit, catalog no. 4368814, Thermo Fisher Scientific). Glyceraldehyde 3-phosphate dehydrogenase (GAPDH) (Thermo Fisher Scientific) was used as an internal control. Expression levels of β-catenin, LEF-1, IL-6, CXCL8, CCL2 and GAPDH were measured by RT-PCR using Taqman^®^ gene expression assays (Table 1) and the StepOnePlus detection system (Thermo Fisher Scientific). The 20 μL reactions were carried out at 60 °C for three min, 95 °C for three min, followed by 40 cycles of 95 °C for 15 s and 60 °C for one min in a 96-well optical RT-PCR plate (Thermo Fisher Scientific). Transcripts were quantified by the comparative ΔΔCT method and represented as fold changes compared to untreated controls.

### 2.8. Western Blot Analyses

Astrocytes were cultured as adherent monolayers in 25 cm^2^ flasks at a density of 4 × 10^6^ cells/flask and allowed to recover for 24 h. Following recovery, cells were treated with IL-1β (20 ng/mL) for 5, 15, 30, and 45 min. Cells were harvested by scraping in sterile ice-cold PBS to avoid alteration of protein expression on the surface of cell membranes. Cell lysate proteins (15 μg) were boiled with 4X NuPAGE lithium dodecyl sulfate loading sample buffer at 100 °C for 5–10 min, resolved by Bolt 4–12% bis tris gel, and subsequently transferred to nitrocellulose membranes using i-Blot (Life Technologies, Carlsbad, CA, USA). The membranes were incubated with antibodies against β-catenin (Cell Signaling Technology, Inc.), GSK-3β (Cell Signaling Technology, Inc.), p-GSK-3β (Cell Signaling Technology, Inc.), or GAPDH (Santa Cruz Biotechnology, Dallas, TX, USA) overnight at 4 °C, washed and then incubated with anti-mouse IgG or anti-rabbit IgG conjugated to horseradish peroxidase (1:10,000, Bio-Rad, Hercules, CA, USA) for 2 h at room temperature. The membrane was then developed with SuperSignal West Femto Substrate (Thermo Fisher Scientific) and imaged in a Flourochem HD2 Imager (ProteinSimple, San Jose, CA, USA). Densitometric analyses were performed using AlphaView software (V3.4.0.0, ProteinSimple).

### 2.9. WES Analyses

Cytoplasmic and nuclear lysates were isolated using a nuclear and cytoplasmic extraction kit according to the manufacturer’s methods (NE-PER, Thermo Fisher Scientific). Protein concentrations were determined by the BCA protein assay kit (Thermo Fisher Scientific) according to the manufacturer’s instructions. Protein (0.4 mg/mL) was loaded into WES capillaries, and columns were probed with antibodies for lamin A/C (mouse, catalog no. NBP-19324, lot no. 41815; WES 1:50, Novus Biologicals, Littleton, CO, USA), β-catenin, NF-κB, and GAPDH. The levels of specific proteins were determined by the WES capillary protein detection system (ProteinSimple) according to the manufacturer’s directions.

### 2.10. Statistical Analysis

Statistical analyses were performed using GraphPad Prism 8.3.1 (GraphPad software, San Diego, CA, USA). All numerical data were taken as mean ± standard error of the mean (SEM). All data were analyzed using student t-test, one-way, or two-way analysis of variance (ANOVA) with Tukey or Fisher’s least significant difference post-hoc tests for pair-wise comparisons. Differences were considered statistically significant at *p* ≤ 0.05.

### 2.11. Data Availability

The R/G-HIV-1 transcriptome data used for the Ingenuity Pathway Analysis (IPA) (version 01-14, Qiagen, Germantown, MD, USA) in this publication have been deposited in NCBI’s gene expression omnibus (GEO) and are accessible through GEO series accession number GSE141756 [47].

## 3. Results

### 3.1. IL-1β and TNF-α Induce Wnt/β-Catenin and NF-κB Signaling in Astrocytes

HIV-1 infection of macrophages and microglia in the CNS leads to cellular activation and secretion of neurotoxic cytokines such as IL-1β and TNF-α [10,11]. These cytokines activate astrocytes and play a key role in astrocyte-associated neuronal injury and HAND pathogenesis. To investigate how IL-1β and TNF-α affect Wnt/β-catenin and NF-κB signaling during HIV-associated neuroinflammation, human astrocyte cultures were exposed to the pro-inflammatory cytokines IL-1β (20 ng/mL) and TNF-α (50 ng/mL) for 8 h either alone or in combination. As a key member of the Wnt/β-catenin pathway, β-catenin mRNA levels were analyzed by RT-PCR. Both IL-1β and TNF-α significantly increased β-catenin transcription (Figure 1A). Levels of LEF-1, a key transcription factor and downstream modulator of the Wnt/β-catenin pathway, were also measured. LEF-1 mRNA levels were elevated by at least 80% regardless of treatment (Figure 1B). These cytokines also significantly increased the NF-κB transcription (Figure 1C). However, combined treatment did not further amplify gene expression of any factors tested. Together, these data show that IL-1β and TNF-α induce LEF-1 and β-catenin expression, suggesting activation of β-catenin signaling in human astrocytes, concurrent with NF-κB signaling. 

### 3.2. β-Catenin Differentially Regulates LEF-1 and NF-κB Transcription in IL-1β and TNF-α-Activated Astrocytes

Crosstalk between Wnt/β-catenin and NF-κB signaling during inflammation in various peripheral cell types has been well summarized by Ma et al. [41]. However, the cross regulation between these two pathways has never been studied in human astrocytes in the context of HIV-associated neuroinflammation. Thus, to assess the crosstalk between Wnt/β-catenin and NF-κB signaling pathways, β-catenin was transiently knocked down in primary human astrocytes, and responses to the HAND-relevant cytokines IL-1β and TNF-α were tested (Figure 2). β-catenin specific siRNA decreased the mRNA levels of β-catenin by approximately 90% with or without IL-1β/TNF-α treatment (Figure 2A) when compared to mock and siCon. The transient knockdown of β-catenin was also evident at protein level (Figure 2D). While LEF-1 transcriptional changes were larger in transfected astrocytes than untransfected astrocytes, silencing β-catenin diminished IL-1β- and TNF-α-induced LEF-1 transcription. Yet, basal LEF-1 transcription was unchanged (Figure 2B). However, silencing β-catenin and LEF-1 did not affect NF-κB transcription (Figure 2C). Together, these data indicate that the Wnt/β-catenin pathway does not directly regulate NF-κB transcription in human astrocytes during HIV-associated inflammation. 

### 3.3. β-Catenin Regulates Wound Healing and Astrocyte Proliferation

Wnt/β-catenin canonical pathway regulates cell proliferation, cell mobility and differentiation. To investigate the role of β-catenin signaling in astrocyte proliferation and migration following injury, either no siRNA (Mock), non-targeting siRNA (siCon) or siRNA specific for β-catenin (siβ-catenin) transfected astrocytes were plated, and a scratch was made following recovery. Wound sites were observed at defined time points to measure astrocyte migration. No difference in astrocyte migration was seen at 8 h in all conditions (Figure 3A1, B1, C1). However, β-catenin knockdown significantly reduced the migration of astrocytes to the injury site at 24 and 36 h (Figure 3C2 and C3, respectively) compared to mock and siCon transfected astrocytes (Figure 3A2, B2 and A3, B3, respectively). Scratched monolayers were completely closed after 48 h in both mock and siCon transfected astrocytes (Figure 3A4 and B4, respectively), leaving minor gaps in the wound of siβ-catenin transfected astrocytes (Figure 3C4).

Canonical Wnt/β-catenin pathway regulates the expression of several transcription factors such as c-myc, cyclin D1 and survivin that mediate cell proliferation [48]. To study the effects of β-catenin pathway on reactive astrogliosis, particularly cell proliferation in vitro, we measured cyclin D1 mRNA levels and BrdU incorporation in human astrocytes following β-catenin knockdown. Significant reductions in cyclin D1 mRNA levels were observed (Figure 3E) following β-catenin knockdown. Simultaneously β-catenin knockdown also resulted in significantly reduced BrdU incorporation, which is a direct measure of cell proliferation (Figure 3F). Together, these data indicate that β-catenin knockdown impairs astrogliosis, i.e., migration and proliferation of astrocytes during injury.

### 3.4. β-Catenin Negatively Regulates Astrocyte IL-6 Expression

To determine if β-catenin regulates neuroinflammation, siRNA transfected astrocytes were treated with IL-1β and TNF-α, and levels of CXCL8, CCL2 and IL-6 were measured at both mRNA and protein levels. IL-1β and TNF-α treatment significantly increased CXCL8 and CCL2 expression both at mRNA (data not shown) and protein levels (Figure 4A and B, respectively). Knockdown of β-catenin did not affect the protein levels of either CXCL8 or CCL2. Treatment with IL-1β and TNF-α also significantly increased IL-6 mRNA and protein levels (Figure 5A and B, respectively). However, IL-1β treatment increased IL-6 to a greater magnitude than TNF-α, and no synergistic effect of the combination treatment was apparent. Moreover, β-catenin knockdown significantly increased IL-1β-induced IL-6 expression alone and in combination with TNF-α, but did not significantly alter TNF-α-induced IL-6. To verify the opposite, the effects of β-catenin activation were also evaluated. Human astrocytes were pre-treated with BIO, a GSK-3β inhibitor that prevents proteasomal-mediated degradation of β-catenin, makes it readily available to translocate into the nucleus, and activates the canonical Wnt pathway. Pre-treated or untreated astrocytes were stimulated with IL-1β and TNF-α, and supernatants were assayed for IL-6 levels by ELISA (Figure 5C). While TNF-α minimally increased IL-6 levels, β-catenin stabilization did not affect TNF-α-induced IL-6 expression. However, in the context of IL-1β alone and in combination with TNF-α, β-catenin stabilization significantly decreased IL-6 secretion (Figure 5C). Together the data indicate that β-catenin negatively regulates IL-6 expression in human astrocytes when exposed to IL-1β alone or in combination with TNF-α.

### 3.5. Stabilization of β-Catenin Increases Nuclear Localization of NF-κB

Wnt/β-catenin signaling involves inhibition of GSK-3β kinase activity by phosphorylation at serine 9, permitting β-catenin accumulation and subsequent expression of Wnt target genes. Since GSK-3β activity is inhibited by phosphorylation at serine 9, we investigated GSK-3β phosphorylation following IL-1β stimulation. Astrocytes were treated with 20 ng/mL of IL-1β for various time points up to 45 min. IL-1β treatment significantly increased GSK-3β phosphorylation at 15 min through 45 min (Figure 6A,B). Data indicate that IL-1β increased the inhibitory phosphorylation of GSK-3β at serine 9, which promotes β-catenin stabilization mechanism in astrocytes. 

To determine if activation of the Wnt/β-catenin signaling pathway regulates the nuclear localization of NF-κB, cells were treated with BIO, cytosolic and nuclear fractions of cell lysates were analyzed for β-catenin and NF-κB levels using simple WES. β-catenin stabilization slightly decreased cytosolic levels of both β-catenin and NF-κB (Figure 6C and D, respectively), which were coupled with increased translocation of both proteins into the nucleus (Figure 6E,F). Together our data indicate that activating the canonical Wnt/β-catenin signaling pathway significantly increased nuclear translocation of NF-κB in human astrocytes.

### 3.6. Molecular Interactions between GSK-3β, β-Catenin, NF-κB and IL-6 during HIVE and HAD

To examine the interrelationships in the context of HIV-infected astrocytes, we accessed a previously published dataset that quantified mRNA expression levels in Red/Green HIV + astrocytes [47]. Expression levels of the molecules of interest, GSK-3β, β-catenin, NF-κB, and IL-6 from the Red/Green-HIV-1 infected human astrocyte data were overlaid with HIV encephalitis (HIVE) and HIV-associated dementia (HAD) disease and biological function using IPA pathway explorer (Figure 7). This analysis indicates multiple pre-existing interactions between the molecules of interest as well as how these molecules interact and affect other molecules implicated in HIVE and HAD. Aligning with the IPA data knowledge, the illustration shows the upregulation of various neuroinflammatory cytokines such as CCL8, CCL2 and IL-6. The illustration also shows that β-catenin affects IL-6 expression both directly and indirectly, while NF-κB is connected to IL-6, CXCL8 and CCL2 as per the IPA curated knowledge. However, our studies demonstrated for the first time that β-catenin negatively regulates IL-6 expression in activated human astrocytes.

## 4. Discussion

In this study we investigated the implications of the Wnt/β-catenin signaling pathway in regulating astrocyte function in the context of HIV-associated neuroinflammation and gliosis. HIV-associated inflammation, IL-1β and TNF-α alone and in combination, strongly induced the expression of proinflammatory cytokines including CCL2, CXCL8 and IL-6 and showed significant activation of both the Wnt/β-catenin and NF-κB signaling pathways in primary human astrocytes. Knockdown of β-catenin impaired astrocyte proliferation and migration but only affected one of the three proinflammatory responses of astrocytes that we examined. IL-1β-induced IL-6 levels were increased by β-catenin knockdown and decreased with BIO, a β-catenin stabilizer, while CCL2 and CXCL8 remained unaffected by β-catenin knockdown. To better understand this relationship, we examined the crossroads between β-catenin and NF-κB pathways. While NF-κB levels were not affected by β-catenin knockdown, β-catenin stabilization increased translocation of both β-catenin and NF-κB p65 into the nucleus in the absence of any other inflammatory stimuli. Together, these data strongly indicate crosstalk between these two pathways in astrocytes during injury and neuroinflammation, wherein reactive astrocyte responses including migration and proliferation were augmented by β-catenin signaling, while the IL-6 response was mitigated. These findings suggest that β-catenin plays an anti-inflammatory role in activated human astrocytes during neuroinflammatory pathologies such as HAND (Figure 8).

At early stages of CNS insults, astrocytes migrate and form barriers at the injured site to restrict the spread of infectious agents or inflammatory cells [49,50]. The β-catenin pathway is a well-known signaling cascade that is involved in migration and the proliferation of several cancers. Initially, we investigated if injury-induced β-catenin signaling contributes to reactive astrocyte phenotype, i.e., proliferation and migration. Following injury, we identified knockdown of β-catenin altered migration of astrocytes to the wound site. The dynamics of β-catenin signaling and its regulation of cell cycle mediators, including cyclin D1 and c-myc expression, are well recognized [48,51]. We also demonstrated β-catenin regulation of cyclin D1 expression in astrocytes, which is essential for proliferation. Taken together, these findings indicate the significance of β-catenin in the proliferative and migration capacity of reactive astrocytes during neurodegenerative diseases.

While HIV-1 predominantly replicates in microglia and macrophages in the CNS, restrictively infected astrocytes were recently shown to be a reservoir that is capable of reinfecting the periphery [52]. Infected macrophages and microglia are the major source of cytokines such as IL-1β, TNF-α and IFN-γ. Apart from the intrinsic upregulation of NF-κB transcription during HIV-1 infection, macrophage-derived inflammatory cytokines, IL-1β and TNF-α, are known to activate NF-κB signaling and induce a reactive astrogliosis phenotype in astrocytes [53,54]. Our recent studies demonstrate that latently infected HIV+ astrocytes resemble actively infected astrocytes and take on a proinflammatory phenotype [47]. 

Reactive astrocytes release a wide variety of inflammatory mediators such as cytokines and chemokines that are both neuroprotective (cytokines such as IL-10 and transforming growth factor-β) and neurotoxic (such as IL-1β, CXCL8, CCL2 and TNF-α) in nature [49]. IL-1β is also known to activate the phosphoinositide 3-kinase (PI3K)/Akt/GSK-3β pathway in several cell types, such as hepatocytes, epithelial cells, airway and colonic smooth muscle cells [55,56]. Inhibitory serine-phosphorylation is the most frequent mechanism that regulates the activity of GSK-3β. Activated Akt, the product of the PI3K pathway, inactivates GSK-3β through phosphorylation, which leads to activation of canonical Wnt/β-catenin signaling. Our studies showed that HIV-associated inflammatory stimuli IL-1β and TNF-α significantly upregulated Wnt/β-catenin signaling and its downstream regulators in human astrocytes. This activation could be due to IL-1β and TNF-α-induced GSK-3β phosphorylation.

Crosstalk between NF-κB signaling and canonical Wnt/β-catenin signaling during inflammation has been extensively studied in the periphery, and has shown both positive and negative associations. Studies have demonstrated modulation of inflammatory responses via interaction of Wnt/β-catenin signaling with NF-κB in murine hepatocytes [57] and positive regulation of NF-κB activity by β-catenin in human bronchial epithelial cells following treatment with lipopolysaccharide. Proinflammatory cytokines, including IL-6, CXCL8, IL-1β, TNF-α and CCL2, were augmented [58]. In colorectal cancer cells, β-catenin regulates the activity of the NF-κB promoter [59], whereas inhibition of GSK-3β by lithium or siRNA potentiated TNF-induced expression of IL-6 and CCL2 in human microvascular endothelial cells [60]. 

Robinson et al. recently demonstrated how the Wnt/β-catenin pathway discordantly regulates IL-6 expression in naïve astrocytes [61]. Robust β-catenin expression decreased IL-6 promoter activity by interacting with T-cell factor (TCF)/LEF transcription factors, independent of NF-κB. However, they also showed that β-catenin augmented NF-κB and CCAAT/enhancer binding protein β expression to boost IL-6 expression [61]. In our studies, neither β-catenin knockdown nor stabilization with BIO altered total NF-κB p65 levels (data not shown). Yet, NF-κB translocation into the nucleus in the absence of astrocytes activation was augmented in concordance with their observations. Further, β-catenin knockdown did not significantly alter basal IL-6 expression in our primary human astrocytes, which was almost undetectable compared to activated levels. While in IL-1β and TNF-α activated astrocytes, the effects of β-catenin on IL-6 levels were significant. Both IL-1β and TNF-α have been shown to activate protein kinase C, which also plays a primary role in the stimulation of IL-6 gene expression in human astrocytes [62]. In this context, β-catenin-associated regulation of astrocyte responses to HIV and neuroinflammation are complex and likely dependent upon the type and gliosis status of these multi-functional cells.

Elevated plasma IL-6 levels have been associated with increased mortality and morbidity in people living with HIV even with effective retroviral suppression [63,64,65]**,** making it a hallmark for HIV-associated inflammation. Early in the epidemic, IL-6 reportedly increased HIV-1 replication in several cell types such as PBMC, monocyte-derived macrophages, and U1 latently infected cells derived from U937 cells [66,67]. This pro-inflammatory cytokine induces B cell differentiation into immunoglobulin producing plasma cells and co-activates various subsets of T cells to mediate immunity. However, during HIV infection, IL-6 may also hinder the immune system by increasing memory T cell turnover and promoting thymic involution impairing T cell reconstitution [68]. In fact, HAND patients show elevated IL-6 levels in cerebrospinal fluid [69,70,71], which are associated with HIV compartmentalization and cognitive impairment. Activation of Wnt/β-catenin signaling, specifically transcription factors LEF-1/4 are known restrictive factors of HIV-1 replication and transcription in astrocytes [27]. NF-κB-induced IL-6 expression in astrocytes was shown to indirectly mediate HIV-associated complement 3 expression to promote neuroinflammation and cognitive impairment [39]. HIV-1 viral proteins such as glycoprotein (gp)120 or negative regulatory factor (Nef) treatment lead to IL-6 expression in human astrocytes, which is mediated by the NF-κB pathway [72,73]. Thus, astrocyte regulation of IL-6 production is an important aspect of HAND pathology and further investigations into these phenomena are warranted. 

## Figures and Tables

**Figure 1 biomedicines-08-00479-f001:**
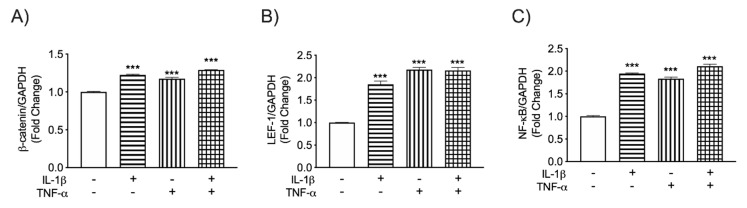
β-catenin, LEF-1 and NF-κB mRNA levels in IL-1β and TNF-α treated human astrocytes. Astrocytes were treated with the HIV-associated cytokines interleukin (IL)-1β (20 ng/mL, horizontal line pattern) and tumor necrosis factor (TNF)-α (50 ng/mL, vertical line pattern) alone or in combination (graph check pattern). Total RNA was isolated 8 h post-treatment. (**A**) β-catenin, (**B**) lymphoid enhancer-binding factor (LEF)-1 and (**C**) nuclear factor (NF)-κB mRNA levels were determined by real-time (RT-PCR). Data display mRNA fold changes compared to untreated controls. Glyceraldehyde 3-phosphate dehydrogenase (GAPDH) was used as a normalizing control. Data cumulative data from three individual astrocyte donors. Statistical analyses were performed using one-way ANOVA with Tukey’s post-hoc test for multiple comparisons (*** *p* < 0.001, compared to an untreated control).

**Figure 2 biomedicines-08-00479-f002:**
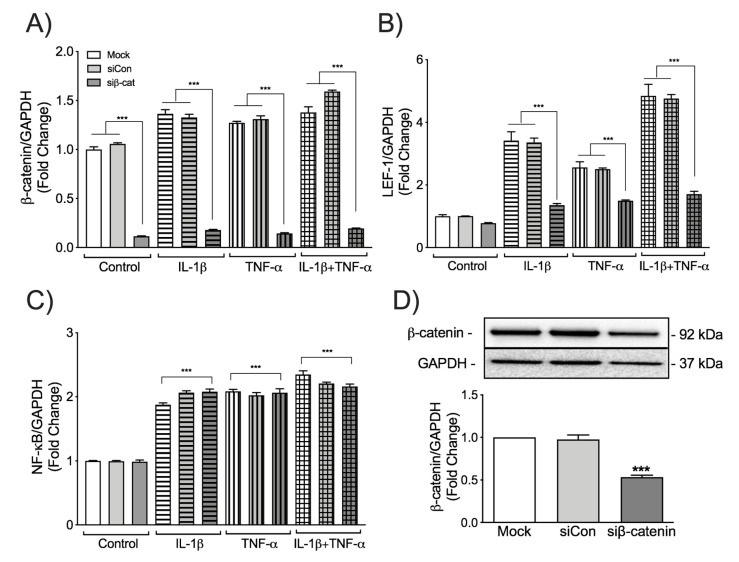
The effects of β-catenin knockdown on LEF-1 and NF-κB induction in activated human astrocytes. Astrocytes were transfected with either no siRNA (Mock, white bars), non-targeting siRNA (siCon, light grey bars) or siRNA specific for β-catenin (siβ-cat, dark grey bars). Astrocytes were then treated with the HIV-associated neurocognitive disorders (HAND)-relevant cytokines IL-1β (20 ng/mL, horizontal pattern) and TNF-α (50 ng/mL, vertical pattern), alone or in combination (graph check pattern). Total RNA was isolated 8 h post-treatment. (**A**) β-catenin, (**B**) LEF-1 and (**C**) NF-κB mRNA levels were determined by RT-PCR. Data display mRNA fold changes compared to untreated controls. GAPDH was used as a normalizing control. Data are cumulative from three individual astrocyte donors. Statistical analyses were performed using one-way ANOVA with Tukey’s post-hoc test for multiple comparisons (*** *p* < 0.001 when compared to Mock or siCon within the treatment). Total protein was isolated 24 h post-treatment and (**D**) probed for β-catenin. β-catenin levels were quantified by densitometry. GAPDH was used as a loading control. Quantification represents cumulative data from three individual astrocyte donors (*** *p* < 0.001 when the treatment was compared to Mock or siCon).

**Figure 3 biomedicines-08-00479-f003:**
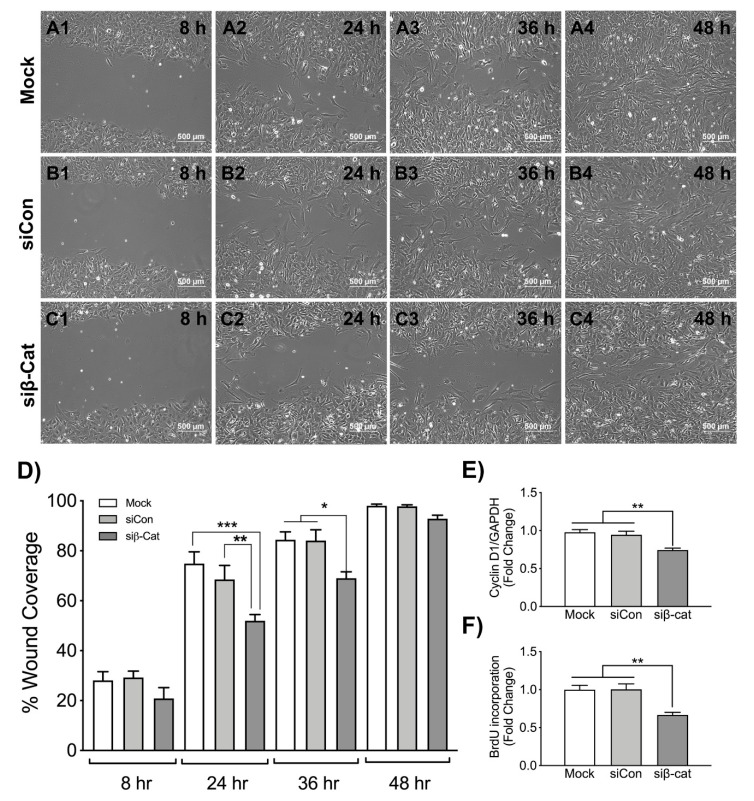
Wound healing and cell proliferation following injury in siβ-catenin transfected astrocytes. Astrocytes were transfected with either no siRNA (Mock, white bars), non-targeting siRNA (siCon, light grey bars) or siRNA specific for β-catenin (siβ-cat, dark grey bars). Post recovery astrocyte monolayers were scratched with a sterile 10 μL pipette tip, and migration of (**A1**–**A4**) mock, (**B1**–**B4**) siCon, and (**C1**–**C4**) siβ-cat transfected astrocytes to the wound site was monitored for 48 h. Phase contrast images captured at 8, 24, 36 and 48 h were shown. Data are representative of three individual donors. Quantified wound healing assay data (**D**) represent an average remaining wound area ± SEM from four individual astrocyte donors at each timepoint. Statistical analyses were performed using one-way ANOVA with Tukey’s post hoc-test for multiple comparisons (* *p* < 0.05, ** *p* < 0.01, *** *p* < 0.001 when the treatments were compared to either Mock or siCon at a given time point). As a measure of astrocyte proliferation and wound healing levels of cyclin D1 were quantified. Total RNA was isolated, and (**E**) cyclin D1 mRNA levels were determined by RT-PCR. GAPDH was used as a normalizing control. Data display fold changes in mRNA levels compared to Mock transfected cells. (**F**) In parallel, 48 h post-transfection, astrocytes were incubated with 10 μM bromodeoxyuridine (BrdU) for 24 h. The graph represents average fold change ± SEM in BrdU incorporation compared to Mock as a measure of cellular proliferation. Data are representative of three individual astrocyte donors. Statistical analyses were performed using one-way ANOVA with Tukey’s post-hoc test for multiple comparisons (** *p* < 0.01, *** *p* < 0.001 when the bars were compared to either Mock or siCon).

**Figure 4 biomedicines-08-00479-f004:**
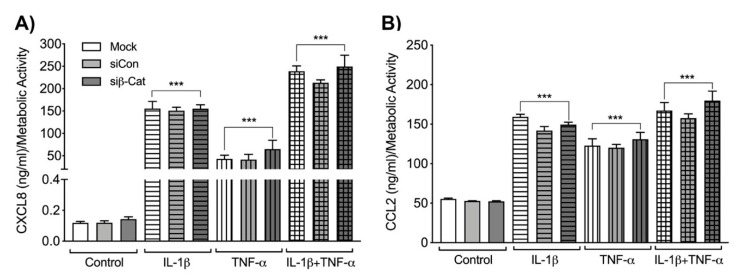
CXCL8 and CCL2 levels following β-catenin knockdown in IL-1β and TNF-α-activated human astrocytes. Astrocytes were transfected with either no siRNA (Mock, white bars), non-targeting siRNA (siCon, light grey bars) or siRNA specific for β-catenin (siβ-cat, dark grey bars). Transfected astrocytes were treated with the HIV-associated cytokines, IL-1β (20 ng/mL, horizontal pattern) and TNF-α (50 ng/mL, vertical pattern), alone or in combination (graph check pattern). Cell culture supernatants were collected 24 h post treatment. (**A**) C-X-C motif chemokine ligand 8 (CXCL8) and (**B**) C-C motif chemokine ligand 2 (CCL2) protein levels were measured by ELISA and normalized to metabolic activity. Data are representative of three individual astrocyte donors. Statistical analyses were performed using one-way ANOVA with Tukey’s post-hoc test for multiple comparisons (*** *p* < 0.001 when the bars were compared to their respective untreated controls).

**Figure 5 biomedicines-08-00479-f005:**
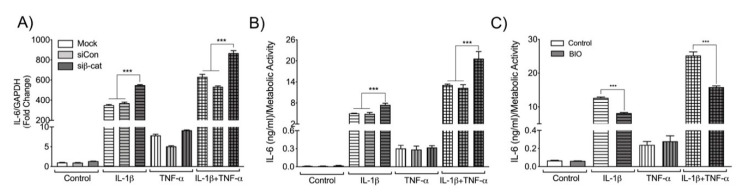
IL-6 levels following β-catenin knockdown in IL-1β and TNF-α-activated human astrocytes. Astrocytes were transfected with either no siRNA (Mock, white bars), non-targeting siRNA (siCon, light grey bars) or siRNA specific for β-catenin (siβ-cat, dark grey bars). Post recovery astrocytes were treated with the HIV-associated cytokines IL-1β (20 ng/mL, horizontal pattern) and TNF-α (50 ng/mL, vertical pattern), alone or in combination (graph check pattern). Total RNA was isolated 8 h post-treatment. (**A**) IL-6 mRNA levels were determined by RT-PCR. Data display mRNA fold changes compared to untreated control. GAPDH was used as a normalizing control. Data are representative of three individual astrocyte donors. Cell culture supernatants were collected 24 h post treatment. (**B**) Protein levels of IL-6 were measured by ELISA and normalized to metabolic activity. Data are representative of three individual astrocyte donors. Human astrocytes were pretreated with a known GSK-3β inhibitor, BIO (500 nM, dark grey bars) or without (Control, white bars) 1 h prior to treatment with IL-1β and TNF-α. Cell culture supernatants were collected 24 h post treatment. (**C**) Protein levels of IL-6 were measured by ELISA and normalized to metabolic activity. Data are representative of three individual astrocyte donors. Statistical analyses were performed using one-way ANOVA with Tukey’s post-hoc test for multiple comparisons. (*** *p* < 0.001 when the bars were compared within the respective treatment groups as indicated).

**Figure 6 biomedicines-08-00479-f006:**
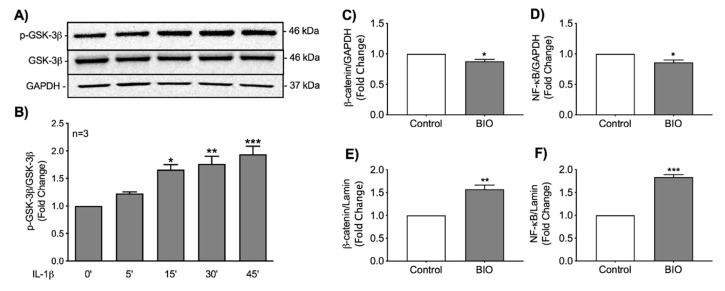
IL-1β-associated GSK-3β phosphorylation and nuclear import of β-catenin and NF-κB in BIO-treated human astrocytes. Primary human astrocytes were treated with IL-1β (20 ng/mL) for 0, 5, 15, 30, and 45 min. Total cell lysates were collected and probed for (**A**) phosphorylated (p)-GSK-3β, total GSK-3β and GAPDH. (**B**) Densitometry analysis was used to calculate the p-GSK-3β to GSK-3β ratio at each timepoint. GAPDH was used as a loading control. Statistical analyses were performed using one-way ANOVA with Tukey’s post-hoc test for multiple comparisons. Data represent average fold change ± SEM from four individual astrocyte donors (* *p* < 0.05, ** *p* < 0.01, *** *p* < 0.001). Human astrocytes were treated with 6-bromoindirubin-3′-oxime (BIO) (500 nM, dark grey bars) or without (Control, white bars). Cytosolic and nuclear fractions of cell lysates were collected 24 h post treatment, (**C**,**E**) β-catenin and (**D**,**F**) NF-κB levels were determined by the WES capillary protein detection system by using specific antibodies. GAPDH and lamin were used as loading controls for cytosol and nuclear fractions, respectively. Data represent average fold change ± SEM from three individual astrocyte donors. Statistical analyses were performed using student’s *t*-test (* *p* < 0.05, ** *p* < 0.01, *** *p* < 0.001).

**Figure 7 biomedicines-08-00479-f007:**
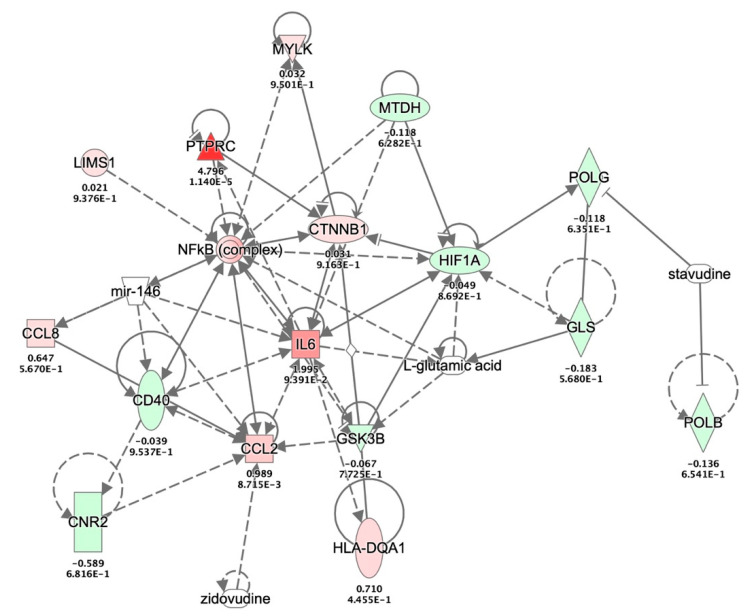
IL-6 network modeling with HIV-associated molecules from HIV+ astrocyte transcriptomes. Network modeling of the molecules from HIV-1 infected astrocyte transcriptome with HIV encephalitis (HIVE) and HIV-associated dementia (HAD) and biological function is generated using Ingenuity Pathway Analysis (IPA) pathway explorer. The generated pathway shows existing interactions between IL-6, β-catenin, and NF-κB during HIVE and HAD. Interactions between proteins are shown by lines, wherein solid lines represent direct interactions, and dotted lines represent indirect interactions. Protein function is indicated by node shape, e.g., enzyme (diamond), cytokine (square), G-protein coupled receptor (rectangle), kinase (triangle), transcription regulator (oval), microRNA (inverted trapezoid) and other (circle). The node color represents the astroglial proteins that are upregulated (red) or downregulated (green) in the R/G-HIV+ human astrocyte transcriptome [47]. Nodes without color represent transcripts/regulators non-RNA effectors like microRNA, antiretrovirals and amino acids. (Abbreviations: CCL2, chemokine ligand 2; CCL8 chemokine ligand 8; CD40, cluster of differentiation 40; CNR2, cannabinoid receptor 2; CTNNB1, β-catenin; GLS, glutaminase; GSK-3B, glycogen synthase kinase 3 beta; HIF1A, hypoxia inducible factor 1 subunit alpha; HLA-DQA1, major histocompatibility complex, class II, DQ alpha1; IL-6, interleukin 6; LIMS1, LIM zinc finger domain containing 1; MTDH, metadherin; MYLK, myosin light chain kinase; NF-κB, nuclear factor kappa-light-chain-enhancer of activated B cells; POLB, DNA polymerase beta; POLG, DNA polymerase gamma; PTPRC, protein tyrosine phosphatase receptor type C)**.**

**Figure 8 biomedicines-08-00479-f008:**
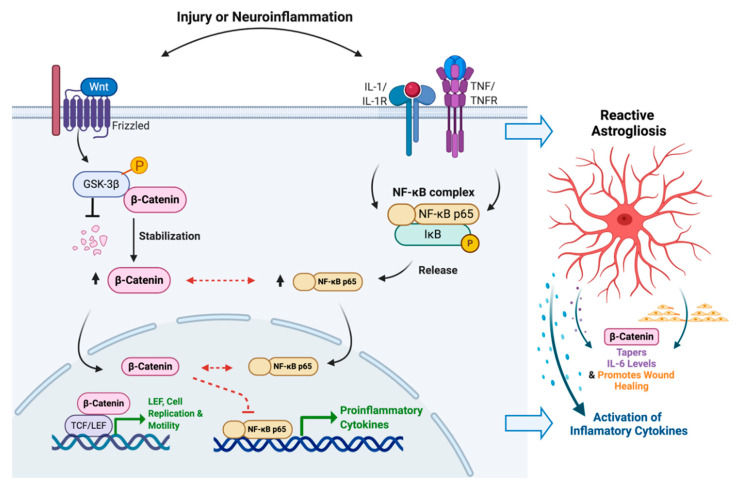
Crosstalk between Wnt/β-catenin and NF-κB signaling during HIV-associated inflammation. The HIV-associated inflammatory cytokines IL-1β and TNF-α upregulated both Wnt/β-catenin and NF-κB signaling in human astrocytes. Stabilization of β-catenin significantly improved translocation of both β-catenin and NF-κB into nucleus (black arrows). Simultaneously, it decreased IL-1β-mediated expression of IL-6 in astrocytes (dashed red line). In spite of the regulatory overlap between Wnt/β-catenin and NF-κB signaling during HIV-associated inflammation, the mechanism (red dashed arrows) of interaction between these two pathways is yet to be decoded in human astrocytes. IL, interleukin; TNF, tumor necrosis factor; R, receptor; GSK, glycogen synthase kinase; NF-κB, nuclear factor of kappa light polypeptide gene enhancer in B-cells; IκB, NF-κB inhibitor; TCF/LEF, T-cell factor/lymphoid enhancer-binding factor. Created with BioRender.com.

**Table 1 biomedicines-08-00479-t001:** Gene expression assay target.

Target	Assay Number (Thermo Fisher)	Dye
Glyceraldehyde 3-phosphate dehydrogenase (GAPDH)	4310884E	(VIC/TAMRA)
β-catenin	Hs00355049_m1	(FAM/MGB)
Lymphoid enhancement-binding factor (LEF-1)	Hs01547250_m1	(FAM/MGB)
Interluekin-6 (IL-6)	Hs00985639_m1	(FAM/MGB)
C-C motif chemokine ligand 2 (CCL2)	Hs002341140_m1	(FAM/MGB)
C-X-C motif chemokine ligand 8 (CXCL8)	Hs00174103_m1	(FAM/MGB)

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
