# Peer review of "β-Catenin Regulates Wound Healing and IL-6 Expression in Activated Human Astrocytes"

_biomedicines, 2020, doi:10.3390/biomedicines8110479_

Round 1

Reviewer 1 Report

This manuscript presents a complex series of experiments analyzing the effect of the beta catenin pathway on a number of primarily stimulatory treatments that mimic the effect of HIV in some CNS infected cells, primarily macrophages and microglia. The authors use astrocytes, a cell type that is known to be infected in some HIV cases, but is neither its primary target nor generally thought to be the principal mediator of HIV associated neurological disease (HAND). Nevertheless this could be an appropriate cell type to study, though the title indicating that they are studying "HIV-associated Inflammation" is a misnomer, since at no point they are infecting astrocytes.

In addition to its potential relevance to HIV, my other main concern with the manuscript is the absence of a 'linear' story. The authors indicate a clear effect of siRNA inhibition of beta catenin pathway on astrocyte migration at some points in their assay. They also indicate that inhibition of this pathway is associated with elevated levels of IL-6 whereas it does not affect other pro-inflammatory cytokines. I am not sure how this helps explain the effects of HIV in the brain nor a potential pathway to inhibit in order to ameliorate these effects. In that regard, Figure 7 does not particularly help figure this out.

In addition:

  1. The investigators are transfecting primary astrocytes. How did they measure the efficiency of this transfection, and particularly any differences among experiments.
  2. Some of the figures (eg Fig 5) should be labelled directly. Instead of Beta catenin negatively regulates IL-6 expression, it should say what it represents: Transfection of siRNA against beta catenin...etc., and leave the interpretation for the description in the text
  3. Why does the effect on migration decrease at 48h? Is this consistent with the time course of beta catenin inhibition?

Author Response

RE: Response to Critiques Reviewer 1

On behalf of the authors, we appreciate your review of our manuscript. We appreciate the opportunity to revise the manuscript to address your concerns.  We have responded to your comments individually below.

This manuscript presents a complex series of experiments analyzing the effect of the beta catenin pathway on a number of primarily stimulatory treatments that mimic the effect of HIV in some CNS infected cells, primarily macrophages and microglia. The authors use astrocytes, a cell type that is known to be infected in some HIV cases but is neither its primary target nor generally thought to be the principal mediator of HIV associated neurological disease (HAND). Nevertheless, this could be an appropriate cell type to study, though the title indicating that they are studying "HIV-associated Inflammation" is a misnomer, since at no point they are infecting astrocytes.

Response:  We apologize for the confusion.  We use the term HIV-associated inflammation to mean inflammation known to occur with HIV infection of the CNS, which we are modeling with IL-1β and TNF-a.  As you describe above, few astrocytes are directly infected with HIV, yet many astrocytes are exposed to a persistent low-level inflammation in the brains of HIV+ individuals. However, you were not alone in this constructive critique of our study. Therefore, we have used more direct descriptions of our treatments with IL-1β and TNF-a in the manuscript and have updated the title appropriately.  Yet, we maintain the HIV-associated context for these studies.

In addition to its potential relevance to HIV, my other main concern with the manuscript is the absence of a 'linear' story. The authors indicate a clear effect of siRNA inhibition of beta catenin pathway on astrocyte migration at some points in their assay. They also indicate that inhibition of this pathway is associated with elevated levels of IL-6 whereas it does not affect other pro-inflammatory cytokines. I am not sure how this helps explain the effects of HIV in the brain nor a potential pathway to inhibit in order to ameliorate these effects. In that regard, Figure 7 does not particularly help figure this out.

Response: We have included Figure 7 to help bridge the gap between these inflammation- associated studies and the transcriptome of HIV+ astrocytes in relation to known molecular associations in HAND brains. The molecular map clearly shows direct and indirect connections between NF-κB, β-catenin and IL-6, while only NF-κB shows associations with other proinflammatory cytokines, such as CXCL8 or CCL2.  Further, the colors in the molecular map indicate the relative change in expression in HIV+ astrocytes as compared to controls. Our study expands these known associations to provide evidence of an anti-inflammatory association between β-catenin and IL-6, also recently described by Lutgen et al. (PLOS Pathogens, 2020) in the absence of inflammation.  The absence of β-catenin, either therapeutically or genetically, it could lead to increased inflammation and worse outcomes in HAND.

In addition:

1.  The investigators are transfecting primary astrocytes. How did they measure the efficiency of this transfection, and particularly any differences among experiments.

Response: This is an excellent question. We measured either mRNA or protein levels in every donor used in these studies. The knockdown efficiency of siβ-catenin was verified in all donors and averaged -88.5 ± 3.6% (± SD) as assayed by real-time PCR and -46.6 ± 3.9% by western blot as described in the methods. The methods (page 3, lines 118-120) were revised to include these assessments. 

2. Some of the figures (eg Fig 5) should be labelled directly. Instead of Beta catenin negatively regulates IL-6 expression, it should say what it represents: Transfection of siRNA against beta catenin...etc., and leave the interpretation for the description in the text.

Response: Thank you for this suggestion; we have revised all figure titles to avoid drawing conclusions for the reader.

3.  Why does the effect on migration decrease at 48h? Is this consistent with the time course of beta catenin inhibition?

Response: The wounds were almost completely healed at this point, thus the minor differences between the controls and siβ-catenin were no longer significant. Knockdown of β-catenin was tested at the RNA and protein levels at 8 h and 24 h post-transfection, and levels remained significantly decreased. However, the loss of effect at 96 hours post-transfection (48 hours post-scratch) could have been influenced by recovery of β-catenin. We do not think that this changes the interpretation or conclusion of the data presented.

We hope that we have satisfactorily addressed many of your concerns and that you will now find our manuscript acceptable for publication.

Reviewer 2 Report

The manuscript submitted by Edara et al (2020) is a very interesting piece of work that describe the effect of knockdown of beta-catenin in astrocytes on the expression of IL-6 after stimulation by IL-1beta alone or in combination with TNF-alpha. The authors observed that only IL-6 levels were negatively regulated by beta-catenin, not proinflammatory cytokines CCL2, or CXCL8. Furthermore, IL-1beta stimulation promotes GSK-3beta phosphorylation and nuclear export of beta-catenin and NF-kappaB. Finally the authors analyzed gene expression patterns from previously published dataset that quantified mRNA expression levels in Red/Green HIV + astrocytes in an effort to link the NF-kappaB and IL-6 nodes. Overall, I find the manuscript well written and the data supportive of most of the conclusions. I do have a few minor concerns which are sketched below.

  1. The words in title “during injury and HIV-associated inflammation” is a little over-stretch as most of the studies were performed without injury or HIV infection. The only “injury” was performed in an in vitro scratch model and IL-1beta and/or TNF-alpha were proxies of “HIV-associated inflammation” or “HAND-related stimuli”. The main thing that relate the authors’ current work to HAND is on the gene expression network analysis in astrocyte infected with R/G HIV+.
  2. The introduction should emphasize a little more on the Wnt/beta-catenin pathways in general and HIV-related pathogenesis in particular. Some sentences in results part could very well be put in the introduction section.
  3. The gene network modeling should be mentioned in abstract.
  4. For the reason mentioned above, the subtitles in Results or Captions in figures, “HAND-relevant stimuli” should be clearly replaced with words like “IL-1beta and TNF-alpha”. For example, “3.1. HAND-relevant stimuli induce Wnt/beta-catenin and NF-kappaB signaling in astrocytes” would be better changed to “3.1. IL-1beta and TNF-alpha induce Wnt/beta-catenin and NF-kappaB signaling in astrocytes.” There are multiple appearances of similar wording in results and in figures.
  5. Discrepancies between Figure 1 and 2. In Figure 1B, IL-1beta stimulation increased the expression level of LEF-1 to 160% compared to control astrocytes. In Figure 2B, IL-1beta stimulation increased the expression level of LEF-1 to about 280% compared to mock siRNA astrocytes. Since the LEF-1 expression level in different experiments varied so big, the comparison in Figure 2B between mock siRNA and beta-catenin siRNA is less reliable than we make it to be.
  6. How were the protein western blots are quantified? What image software was used?

Author Response

RE: Response to Critiques Reviewer 2

The manuscript submitted by Edara et al (2020) is a very interesting piece of work that describe the effect of knockdown of beta-catenin in astrocytes on the expression of IL-6 after stimulation by IL-1beta alone or in combination with TNF-alpha. The authors observed that only IL-6 levels were negatively regulated by beta-catenin, not proinflammatory cytokines CCL2, or CXCL8. Furthermore, IL-1beta stimulation promotes GSK-3beta phosphorylation and nuclear export of beta-catenin and NF-kappaB. Finally the authors analyzed gene expression patterns from previously published dataset that quantified mRNA expression levels in Red/Green HIV + astrocytes in an effort to link the NF-kappaB and IL-6 nodes. Overall, I find the manuscript well written and the data supportive of most of the conclusions. I do have a few minor concerns which are sketched below

Response:  On behalf of the authors, we thank you for your thorough review of our manuscript and your constructive criticisms, which we address below.

  1. The words in title “during injury and HIV-associated inflammation” is a little over-stretch as most of the studies were performed without injury or HIV infection. The only “injury” was performed in an in vitro scratch model and IL-1beta and/or TNF-alpha were proxies of “HIV-associated inflammation” or “HAND-related stimuli”. The main thing that relate the authors’ current work to HAND is on the gene expression network analysis in astrocyte infected with R/G HIV+.

Response: As you surmised, we used the term HIV-associated inflammation to mean inflammation known to occur with HIV infection of the CNS, which we are modeling with IL-1β and TNF-α.  Few astrocytes are directly infected with HIV, yet many astrocytes are exposed to a persistent low-level inflammation in the brains of HIV+ individuals. However, you were not alone in this constructive critique of our study. Therefore, we have used more direct descriptions of our treatments with IL-1β and TNF-α in the manuscript and have updated the title appropriately.

  1. The introduction should emphasize a little more on the Wnt/beta-catenin pathways in general and HIV-related pathogenesis in particular. Some sentences in results part could very well be put in the introduction section.

Response: We appreciate the guidance for improving our introduction and have added a paragraph to emphasize regulation of the β-catenin pathway and its relation to HIV inflammation (page 2, lines 71-84). We reiterated some important points for the readers in the results to help them relate to the context of the experiments and data described.

  1. The gene network modeling should be mentioned in abstract.

Response: Thank you for this suggestion. Lines 39-41 of the abstract now read, “HIV+ human astrocytes show increased IL-6, β-catenin and NF-κB expression levels and are interconnected by regulatory associations during HAND.”

  1. For the reason mentioned above, the subtitles in Results or Captions in figures, “HAND-relevant stimuli” should be clearly replaced with words like “IL-1beta and TNF-alpha”. For example, “3.1. HAND-relevant stimuli induce Wnt/beta-catenin and NF-kappaB signaling in astrocytes” would be better changed to “3.1. IL-1beta and TNF-alpha induce Wnt/beta-catenin and NF-kappaB signaling in astrocytes.” There are multiple appearances of similar wording in results and in figures.

Response: As discussed above, you were not alone in this concern, thus we have thoroughly revised the manuscript to directly describe our treatments and results for IL-1β and TNF-α treatment. 

  1. Discrepancies between Figure 1 and 2. In Figure 1B, IL-1beta stimulation increased the expression level of LEF-1 to 160% compared to control astrocytes. In Figure 2B, IL-1beta stimulation increased the expression level of LEF-1 to about 280% compared to mock siRNA astrocytes. Since the LEF-1 expression level in different experiments varied so big, the comparison in Figure 2B between mock siRNA and beta-catenin siRNA is less reliable than we make it to be.

Response: This was an excellent catch by the reviewer.  We had not directly compared the magnitudes of the LEF-1 responses in non-transfected (Fig. 1B) and transfected cells (Fig. 2B).  It does appear that transfected cells (regardless of condition) had significantly larger LEF expression levels. While we do not understand the exact reason,  we have added a description of this phenomenon to the results section (page 6, lines 229-230).  We do not think that this diminishes the conclusions drawn in Figure 2, panel B as LEF-1 was responsive to siβ-catenin knockdown as compared to the internal controls, which remained comparable across mock, siCON and siβ-catenin.  We have also updated these figures to show cumulative data rather than representative donors, to account for variation between donor astrocyte cultures.

  1. How were the protein western blots are quantified? What image software was used?

Response: We regret this omission in our methods; it now reads, “Densitometric analyses were performed using AlphaView software (ProteinSimple, V3.4.0.0).” (page 5, lines 174-175).

We hope that we have satisfactorily addressed many of your concerns and that you will now find our manuscript acceptable for publication.

Reviewer 3 Report

The manuscript by Borgmann et al discusses the potential link between reactive astrogliosis and HAND. The authors proposed that inflammatory modulators associated with HIV potentiates β-catenine pathway, leading to reactive astrocytic response while alleviating IL-6 expression. Taken together, the authors suggest that components of β-catenine pathway has anti-inflammatory role in astrocytes during HAND. The authors efforts are commendable, and the study adds significant value in understanding the role of β-catenine in astrogliosis.

Comments:

  1. The authors aimed to demonstrate the interrelation between reactive astrogliosis and HAND via β-catenine. From that perspective, it would be useful to demonstrate the mechanistic pathway in HIV infected astrocytes. While it is understood the rarity of HIV-infected fetal astrocytes, have the authors tried to infect the primary astrocytes with HIV-1 isolates?
  2. A graphical representation of the proposed mechanism would be helpful.
  3. Is there any established protocol for experimental method section 2.1? If so, please include the reference.

Author Response

The manuscript by Borgmann et al discusses the potential link between reactive astrogliosis and HAND. The authors proposed that inflammatory modulators associated with HIV potentiates β-catenin pathway, leading to reactive astrocytic response while alleviating IL-6 expression. Taken together, the authors suggest that components of β-catenin pathway has anti-inflammatory role in astrocytes during HAND. The authors efforts are commendable, and the study adds significant value in understanding the role of β-catenin in astrogliosis.

Response: On behalf of the authors, we appreciate your review of our manuscript. We appreciate the opportunity to revise the manuscript to address your concerns.  We have responded to your comments individually below.

Comments:

  1. The authors aimed to demonstrate the interrelation between reactive astrogliosis and HAND via β-catenin. From that perspective, it would be useful to demonstrate the mechanistic pathway in HIV infected astrocytes. While it is understood the rarity of HIV-infected fetal astrocytes, have the authors tried to infect the primary astrocytes with HIV-1 isolates?

Response: The data presented in Figure 7 are from our HIV+ astrocyte studies, which show increased β-catenin, NF-κB and IL-6 levels.  Pure populations of HIV+ astrocytes are sorted by FACS, that yield small cell numbers which are difficult to manipulate. Plus, we do not know if they would survive transfection with siRNAs. However, this is most definitely a study that warrants further investigation!

  1. A graphical representation of the proposed mechanism would be helpful.

Response: Thank you for this suggestion; we have added this as Figure 8, which illustrates what gaps in the mechanism remain.

  1. Is there any established protocol for experimental method section 2.1? If so, please include the reference.

Response: We are currently working with JOVE on a publication of this method. However, we have described it in detail in several publications, three of which are cited.

We hope that we have satisfactorily addressed many of your concerns and that you will now find our manuscript acceptable for publication.

Round 2

Reviewer 1 Report

The manuscript has been considerably improved, particularly by simplifying the presentation to the work actually performed and eliminating direct references to HIV.